# R-spondin signalling is essential for the maintenance and differentiation of mouse nephron progenitors

Valerie PI Vidal[1], Fariba Jian-Motamedi[1], Samah Rekima[1], Elodie P Gregoire[1], Emmanuelle Szenker-Ravi[2], Marc Leushacke[2], Bruno Reversade[2], Marie-Christine Chaboissier[1], Andreas Schedl[1]*

[1]Université Côte d'Azur, Inserm, CNRS, Institut de Biologie Valrose, Nice, France; [2]Institute of Medical Biology, A*STAR, Singapore, Singapore

**Abstract** During kidney development, WNT/β-catenin signalling has to be tightly controlled to ensure proliferation and differentiation of nephron progenitor cells. Here, we show in mice that the signalling molecules RSPO1 and RSPO3 act in a functionally redundant manner to permit WNT/β-catenin signalling and their genetic deletion leads to a rapid decline of nephron progenitors. By contrast, tissue specific deletion in cap mesenchymal cells abolishes mesenchyme to epithelial transition (MET) that is linked to a loss of *Bmp7* expression, absence of SMAD1/5 phosphorylation and a concomitant failure to activate *Lef1, Fgf8* and *Wnt4,* thus explaining the observed phenotype on a molecular level. Surprisingly, the full knockout of LGR4/5/6, the cognate receptors of R-spondins, only mildly affects progenitor numbers, but does not interfere with MET. Taken together our data demonstrate key roles for R-spondins in permitting stem cell maintenance and differentiation and reveal *Lgr*-dependent and independent functions for these ligands during kidney formation.

*For correspondence:
schedl@unice.fr

Competing interests: The authors declare that no competing interests exist.

## Introduction

Nephron endowment is a critical factor for renal health and low number of nephrons have been associated with chronic kidney disease and hypertension (*Bertram et al., 2011*). In mammals, nephrogenesis is restricted to the developmental period and involves a dedicated nephrogenic niche at the most cortical region of the forming kidney that fuels successive rounds of nephron production (*McMahon, 2016*; *Oxburgh, 2018*; *O'Brien, 2019*). The nephrogenic niche consists of three independent progenitor populations that will develop into collecting ducts, stroma (interstitial cells) and nephrons. Nephron progenitor cells (NPCs) form a condensed cap around the tips of the branching ureter. This cap mesenchyme (CM) can be further subdivided into populations that represent cells of progressive differentiation status depending on the position along their migration trajectories around the ureteric tips. Analysis over the last decade have identified at least four different subpopulations: 1) CITED1[+]/SIX2[+] that are considered as 'ground-state' progenitors; 2) CITED1[-]/SIX2[+] progenitors; 3) primed CITED1[-]/SIX2[+]/pSMAD[+] progenitors; 4) WNT4[+]/LEF1[+] pretubular aggregates (PTA). Once engaged, PTAs undergo a mesenchyme to epithelial transition (MET) to form epithelialized renal vesicles, which will further differentiate via comma- and S-shaped bodies into segmented nephrons. This traditional view of a molecular and spatial subdivision of the CM has been challenged by findings that nephron progenitor cells (NPCs) are much more mobile than previously appreciated. Indeed, NPCs have been observed to travel back and forth between caps of independent tips (*Combes et al., 2016*) and between the pretubular aggregate and cap mesenchyme state (*Lawlor et al., 2019*).

**eLife digest** Kidneys filter waste out of the bloodstream to produce urine. Each kidney contains many structures called nephrons which separate the waste from the blood. The number of nephrons in a kidney varies between people, and those with low numbers have a higher risk of chronic kidney disease. Nephrons are formed before birth from a specific group of so-called progenitor cells. Each of these cells can either divide to make others like itself, or it can specialize to make nephron cells. At the end of embryonic kidney development, all the progenitor cells become nephron cells.

Cells that specialize to become part of a nephron first go through a change called a mesenchyme-to-epithelial transition. Epithelial cells move less than mesenchymal cells, and also develop a clear structure where the two ends of the cell adapt to different roles. Evidence suggests that a cell communication process called WNT/β-catenin signaling controls this transition. Yet the details of how this transition is controlled are not fully understood. One way to activate WNT/β-catenin signaling is with R-spondin proteins, which have been found in developing kidneys.

Vidal et al. studied R-spondins during the embryonic development of kidneys in mice. Removing R-spondins stopped the progenitor cells from producing more of themselves and increased the number that died. The R-spondins were also needed for the progenitor cells to specialize as nephron cells through the mesenchyme-to-epithelial transition. Further results revealed that R-spondins activate WNT/β-catenin signaling in these cells, even though the proteins that usually act as R-spondin receptors (called LGR4/5/6) could be removed without affecting the results. This suggests that R-spondins interact with different receptor proteins during kidney development.

These findings highlight the role of R-spondins and WNT/β-catenin signaling in kidney development. Future studies will seek the receptor proteins that R-spondins interact with in kidneys. They may also look to understand how R-spondins balance their different roles in progenitor cells and during cell specialization. These results in mice could also be extended to determine their relevance in human health and disease, including chronic kidney disease, which is responsible for more deaths than breast or prostate cancer.

The transformation of nephron progenitors into epithelialized nephrons is not an entirely cell-intrinsic process, but requires inductive signals from both the ureteric tip and stromal cells (*O'Brien, 2019*). Of particular importance is WNT9b, a molecule released from the branching ureter that induces canonical WNT/β-catenin signalling, stimulates proliferation and thus ensures self-renewal of kidney progenitors. Accordingly, deletion of β–catenin leads to the loss of progenitor cells (*Karner et al., 2011*). However, canonical WNT signalling is also required for MET (*Carroll et al., 2005*) and transient activation of β–catenin in isolated progenitors induces epithelialisation (*Park et al., 2007*; *Kuure et al., 2007*; *Park et al., 2012*). How the balance between progenitor proliferation and differentiation is achieved is not well understood, but experimental evidence suggests that progenitor proliferation requires low levels of β-catenin activity, whereas genes that are involved in MET are activated in the presence of a strong canonical β-catenin signal (*Ramalingam et al., 2018*). In the context of nephrogenesis, a strong β-catenin response appears to rely on the activation of canonical BMP/SMAD pathway, as progenitor cells leaving the niche are positive for pSMAD and deletion of *Bmp7* interferes with MET (*Brown et al., 2013*).

WNT/β-catenin signalling is essential for many organ systems and multiple feedback mechanisms have been identified that control signalling strength at almost every level of this signal transduction pathway. WNT receptor availability at the cell membrane is controlled by RNF43 and ZNRF3, two trans-membrane E3 ubiquitin ligases that induce receptor endocytosis and thus negatively regulate WNT signalling. Their action is counteracted by R-spondins (RSPO1-4), a family of secreted molecules that bind to the G-protein-coupled receptors LGR4/5/6. Binding to LGRs permits R-spondins to interact with RNF43/ZNRF3 and suppress endocytosis of the WNT receptor complex, thus enhancing WNT signalling (*de Lau et al., 2014*).

In this study, we investigated a potential role of the R-spondin/LGR axis in controlling renal stem/progenitor behaviour in vivo. We show that *Rspo1* and *Rspo3* are required to maintain the pool of renal progenitors throughout development by supporting their proliferative capacity and preventing their apoptosis. Moreover, strong *R-spondin* signal is essential to allow nephron progenitors to

engage in differentiation and undergo MET. RSPO1/3 achieve these functions by their ability to activate the WNT/β−catenin signalling pathway, a role that is primarily mediated in an LGR-independent manner.

## Results

### *R-spondins* are dynamically expressed during kidney development

To understand the role of R-spondins during kidney development in mice, we first mapped the expression of the four members of this gene family using qPCR and in situ hybridisation analysis. Although *Rspo2* and *Rspo4* were undetectable in developing kidneys (*Figure 1—figure supplement 1A—source data 1*), *Rspo1* and *Rspo3* could be found as early as E10.5 within SIX2$^+$ renal progenitors (*Figure 1—figure supplements 1B* and *Motamedi et al., 2014*). Interestingly, *Rspo3* marked only a proportion of SIX2 positive cells, suggesting this population to be heterogeneous already at this early age (*Figure 1—figure supplement 1B*). At E14.5, *Rspo1* was detected throughout the CM, PTA, within renal vesicles, and the proximal part of the comma- and S-shaped bodies, but decreased upon podocyte differentiation (*Figure 1A* and *Figure 1—figure supplement 1C*). By contrast, *Rspo3* expression was restricted to uncommitted SIX2+ cells (*Figure 1B* and *Figure 1—figure supplement 1B–C*), and what appeared to be low levels of expression within the cortical stroma (*Figure 1—figure supplement 1C*). Indeed, expression within the most cortical population of stromal cells persisted in animals that carry a CM-specific deletion of *Rspo3* (*Six2:Cre*, *Rspo3$^{fl/fl}$*) (*Figure 1—figure supplement 1C*). Interestingly, by E18.5 *Rspo3* was dramatically reduced in NPCs, but strongly expressed in the cortical stromal compartment (*Figure 1Bii*), indicating a shift of expression towards the stroma. Strong *Rspo3* signal was also detected in stromal cells lining ducts of the renal papilla (*Figure 1Biii*).

### RSPO1 and RSPO3 are essential to maintain the pool of kidney progenitors

Lack of *Rspo1* is compatible with life (*Chassot et al., 2008*) and kidneys isolated from knockout mice showed no discernible abnormalities (data not shown). Mice carrying a constitutive deletion of *Rspo3* die at E9.5 due to placental defects (*Aoki et al., 2007*; *Kazanskaya et al., 2008*). To overcome this early lethality, we employed a conditional allele for *Rspo3* (*Rspo3$^{fl/fl}$*) (*Rocha et al., 2015*) in combination with a range of *Cre*-expressing lines (*Figure 1—figure supplement 2*). Tamoxifen (Tam) induction at E11.5 in presence of the ubiquitously expressed *CAGG:CreER$^{TM}$* driver (*Hayashi and McMahon, 2002*) efficiently abolished *Rspo3* expression 3 days after induction (E14.5) (*Figure 1—figure supplement 1A*) and resulted in a mild reduction of progenitor cells (*Figure 1—figure supplement 3B*). To test whether *Rspo1* and *Rspo3* may act in a functionally redundant manner, we induced deletion at E11.5 and analysed the renal phenotype at E14.5 and E18.5 in double mutant animals (*CAGG:CreER$^{TM}$; Rspo1$^{-/-}$, Rspo3$^{fl/fl}$* - from now on called DM). Depletion was efficient, and no compensatory up-regulation of *Rspo2* or *Rspo4* was detected in double mutant kidneys (*Figure 1—figure supplement 1A*). Macroscopic observation at E18.5 showed severe renal hypoplasia in *R-spondin* DM, when compared to control littermates (*Figure 1C,D*). Hematoxylin and Eosin (H&E) staining of kidneys at E14.5 indicated a reduced nephrogenic zone and a near-complete absence of nephrons (*Figure 1E*). Molecular analysis confirmed a dramatic loss of SIX2$^+$ progenitors and an absence of forming nephrons (*Figure 2A*). The remaining SIX2-positive cells appeared scattered, when compared to the condensed mesenchyme seen in controls. A similar loss of NPCs was observed when *R-spondin* deletion was induced at E10.5, E12.5 or E13.5 indicating a continuous requirement of these signalling molecules for the maintenance of the NPC pool (data not shown). To test whether the reduction of progenitors was caused by defects in proliferation or apoptosis, we next quantified BrdU incorporation and TUNEL staining (*Figure 2B–C*; *Figure 2—source data 1*; *Figure 2—figure supplement 1B,C*). In order to measure early events of R-spondin deletion within progenitors, we allowed the first wave of nephrons to form, TAM-induced CRE activity at E12.5 and analysed samples 2 days thereafter. Quantification of BrdU-labelled SIX2$^+$ progenitors indicated a 40% reduction of proliferation in this compartment (p=0.0003) (*Figure 2B*). In addition, a significant 8.1 fold increase of apoptosis among the SIX2$^+$ cells located in the CM was observed (p=0.0319) (*Figure 2C*).

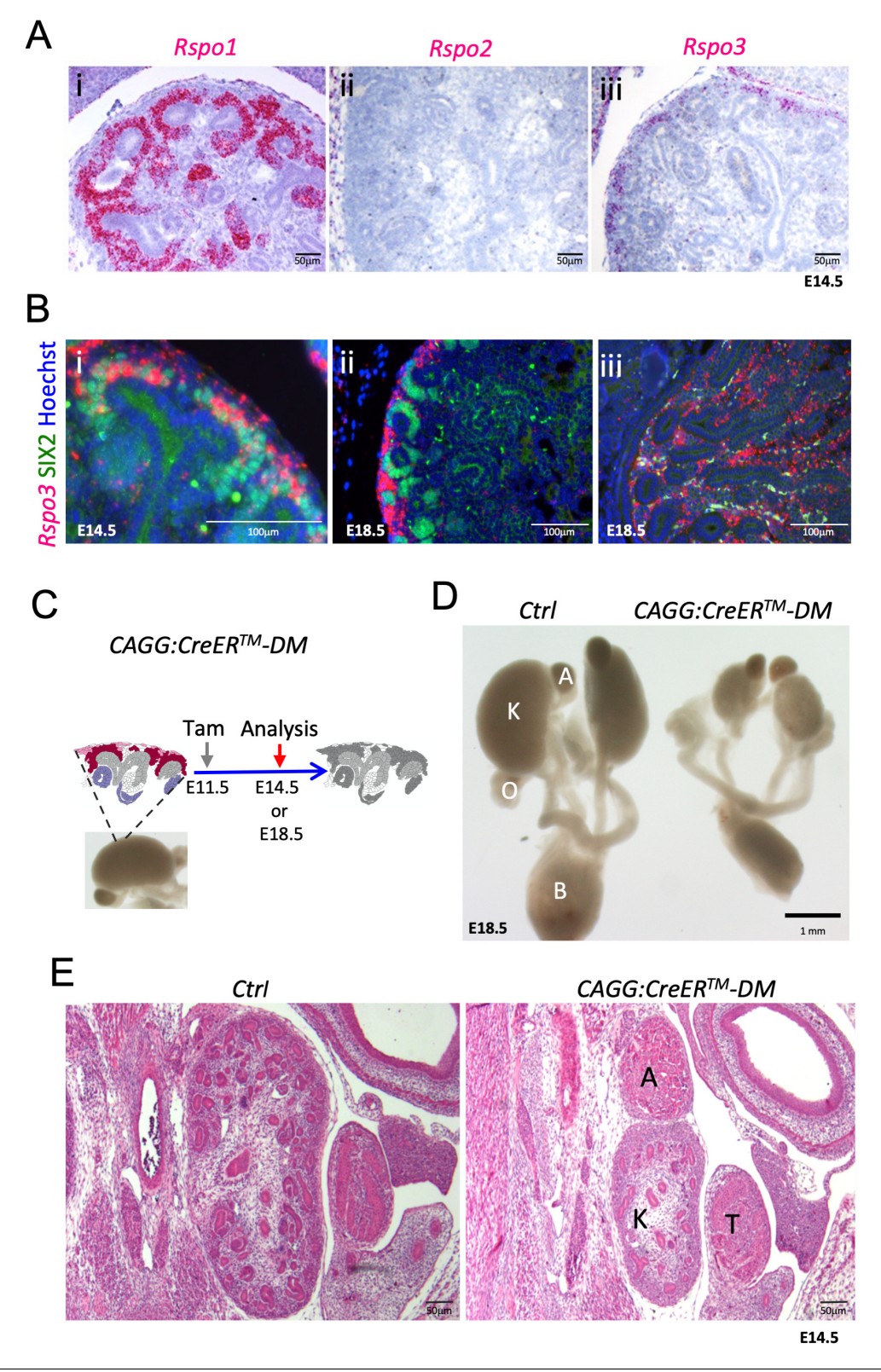

**Figure 1.** *Rspo1* and *Rspo3* are expressed in embryonic kidneys and are required for normal development. (**A**) RNA-Scope analysis demonstrates *Rspo1* (**i**) and *Rspo3* (**iii**) expression in the nephrogenic zone of developing (E14.5) kidneys. (**B**) RNA-Scope analysis followed by immunostaining for the progenitor marker SIX2 reveals a switch from strong *Rspo3* expression within progenitors at E14.5 (**i**) to almost exclusively stromal progenitor

*Figure 1 continued on next page*

*Figure 1 continued*

expression at E18.5 (ii). In addition, strong staining was found within medullary stromal cells (iii). Hoechst stains nuclei in blue (C) Schematic outline of tamoxifen induction for *CAGG:CreER^TM*-mediated deletion leading to a complete loss of *R-spondin* expression in the *Rspo1/Rspo3* double mutant (DM). Colour legend: Purple for Rspo1+, pink for Rspo3+, dark red for Rspo1/Rspo3+ cells. Rspo depleted cells are dark grey, light grey highlights ureteric cells. (D) Macroscopic view of the urogenital system of Control and *CAGG:CreER^TM-DM* embryos dissected at E18.5, (E). Hematoxylin and Eosin (Hand E) staining of E14.5 sections reveals smaller kidneys virtually lacking nephrons. A: adrenal gland, B: bladder, K: kidney, O: ovary, T: testis. CAGG:CreER^TM-DM stands for (CAGG:CreER^TM, Rspo1^{-/-}, Rspo3^{fl/fl}), Ctrl: Control.

The online version of this article includes the following source data and figure supplement(s) for figure 1:

**Figure supplement 1.** *R-spondin* expression analysis during kidney development.
**Figure supplement 1—source data 1.** Quantitative analysis of all R-spondin genes expression in *Rspo1* and *Rspo3*-mutant kidneys.
**Figure supplement 2.** Schematic outline of the different genetic approaches used in this study.
**Figure supplement 3.** *Rspo1* and *Rspo3* are functionally redundant.

As R-spondins are known activators of *Wnt*/β-catenin signalling, we analysed the expression of *Axin2*, a direct downstream target and read-out of canonical β–catenin signalling. Ubiquitous deletion of R-spondins caused a general downregulation of *Axin2* already 2 days after tamoxifen induction, an observation consistent with a loss of WNT/β–catenin signalling (*Figure 2—figure supplement 1D*). Progenitor-specific deletion of β-catenin (*Six2:Cre; Ctnnb1^{fl/fl}*) phenocopied the *Rspo1*/3 DM phenotype with a dramatic reduction (93%; p<0.0001) of SIX2^+ progenitor cells and a near-complete block of nephron formation as indicated by an almost complete absence of the nephron specific marker JAG1 (*Figure 2D and E*; *Figure 2—source data 1*). Taken together these data indicate a requirement of R-spondins for nephron progenitor maintenance via the activation of canonical β-catenin signalling.

## Stromal *Rspo3* maintains nephron progenitors during late stages of kidney development

*Rspo3* is produced by both, stromal and nephron progenitors, but at later stages expression shifts to a predominantly stromal expression (compare *Figure 1Bi* with *Figure 1Bii*). To evaluate the contribution of stromally derived RSPO3, on kidney development, we specifically depleted its expression in this compartment using the constitutively active *Foxd1:Cre* line. In this context, progenitor cells were the only source of RSPO3 (*Figure 3A* and *Figure 1—figure supplement 2*). At E14.5, *Foxd1:Cre-DM* kidneys appeared normal and showed a comparable number of SIX2^+ cells per section when compared to control littermates (p=0.7887; *Figure 3C*; *Figure 3—source data 1* and data not shown). By contrast, at E18.5 mutant kidneys were hypoplastic (*Figure 3B*) and displayed a reduction of the nephrogenic zone associated with a significant loss of SIX2+ progenitor cells (p=0.0167) (*Figure 3D–F*; *Figure 3—source data 1*). We conclude that continuous expression of *Rspo3* from the stromal compartment is required to maintain progenitor cells at later stages of kidney formation.

## RSPO1 and RSPO3 are required for mesenchyme-to-epithelial transition

To test the role of progenitor-released *R-spondins*, we used a constitutively active progenitor specific Cre-line (*Six2:Cre*) to delete expression of *Rspo3* in this cell compartment. Consequently, in *Rspo1^{-/-}; Six2:Cre; Rspo3^{fl/fl}* mice (*Six2:Cre-DM*), stromal *Rspo3* is the only remaining R-spondin (*Figure 4A* and *Figure 1—figure supplement 2*). At E18.5 *Six2:Cre-DM* embryos had hypodysplastic kidneys that histologically displayed defects in nephrogenesis and a complete absence of glomeruli (*Figure 4B*). Histological and immunofluorescent analysis of E14.5 kidneys confirmed this observation and revealed the persistence of SIX2+ progenitors albeit at slightly lower numbers (*Figure 4C* and *Figure 4—figure supplement 1B*). Importantly, staining for JAG1 and WT1 demonstrated a complete absence of epithelial nephrons and podocytes, respectively, indicating an essential role for *R-spondin1/3* in nephron differentiation.

The above phenotype is reminiscent of defects seen in mutants for *Wnt9b*, a signalling molecule that is released from the branching ureter. WNT9b activity has been shown to activate two classes of

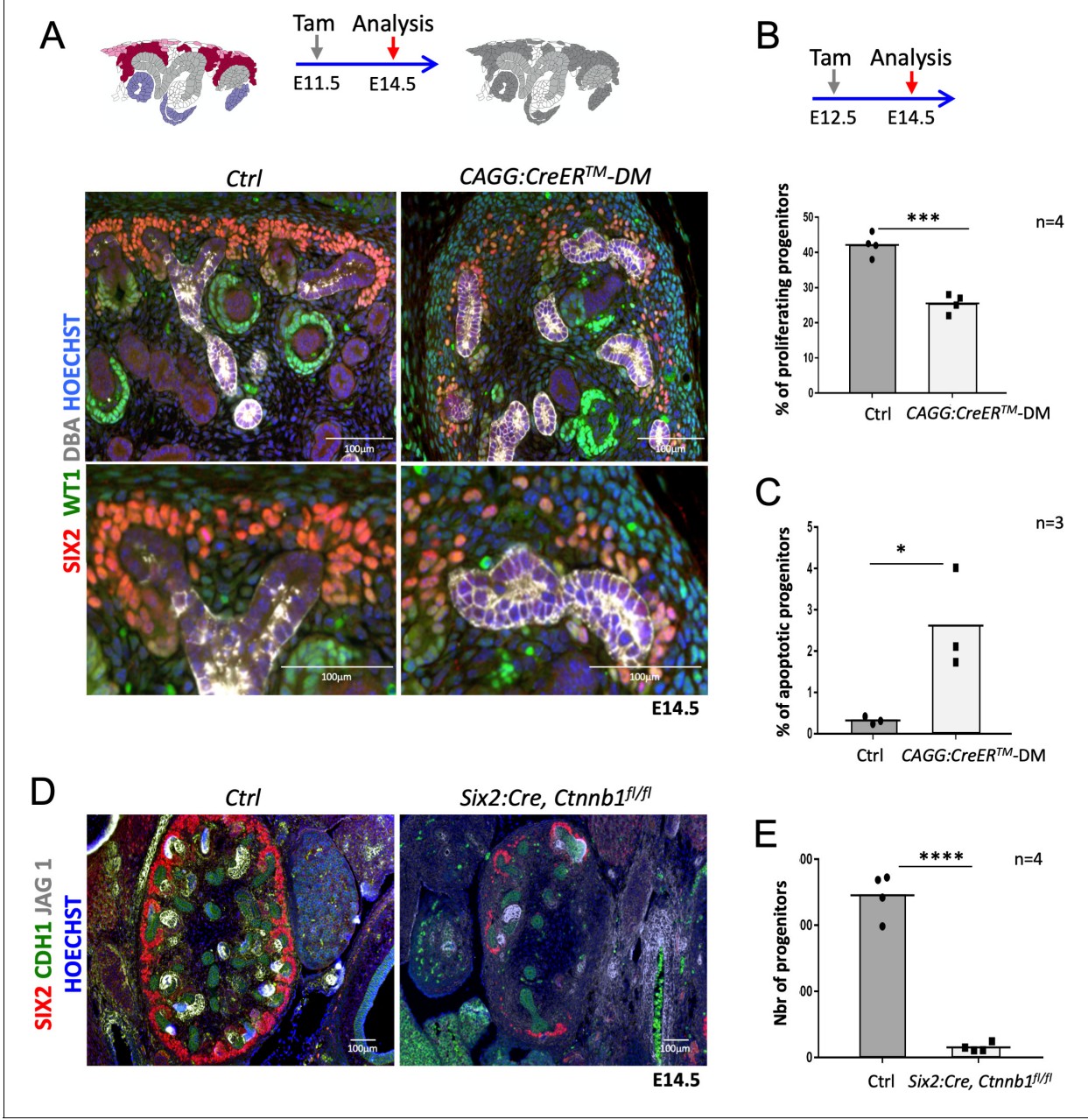

**Figure 2.** R-spondins are required for renal progenitor maintenance. (**A**) Immunofluorescent analysis at E14.5 (induced at E11.5) reveals loss of SIX2+ progenitor cells and nascent nephrons (comma or S-shaped bodies) in *CAGG:CreER^{TM}-DM* embryos. (WT1 = green; SIX2 = red; DBA = white; Hoechst = blue). Colour legend for the cartoon: Purple label Rspo1+, pink Rspo3+, dark red Rspo1/Rspo3+ cells. Rspo depleted cells are dark grey, light grey highlights ureteric cells. (**B**) Quantification of BrdU-labelled SIX2+ progenitors performed on four embryos (n = 4) demonstrates a significant reduction of proliferation 2 days after *Rspo3* deletion. See *Figure 2—source data 1* (**C**) TUNEL analysis reveals a dramatic increase in apoptosis (n = 3 embryos for each genotype, two litters). (**D**) Progenitor specific deletion of β-catenin (*Six2:Cre; Ctnnb1^{fl/fl}*) results in the loss of progenitor cells at E14.5 (SIX2 = red; CDH1 = green; JAG1 = white). (**E**). Quantification of SIX2+ progenitors (n = 4 embryos for each genotype isolated from two litters). See *Figure 2—source data 1*. Columns are means ± SEM with p<0.05 (*), p<0.01 (**), p<0.001 (***), p<0.0001 (****). One black dot = average value for one control embryo, one black square = average value for one *CAGG:CreER^{TM}-DM* embryo.

The online version of this article includes the following source data and figure supplement(s) for figure 2:

**Source data 1.** Progenitors quantification in*CAGG:CreER^{TM}-DM*and (*Six2:Cre;Ctnnb1^{fl/fl}*) mutant kidneys.

**Figure supplement 1.** Proliferation and apoptosis analysis.

β–catenin dependent genes: Class II genes that require lower levels of β–catenin signalling and are found predominantly in nephron progenitors; Class I genes such as *Wnt4* that depend on strong β–catenin activation and are highly expressed in pretubular aggregates (*Karner et al., 2011*; *Ramalingam et al., 2018*). In situ hybridisation (ISH) analysis revealed that loss of R-spondins did not impact *Wnt9b* expression (*Figure 4Di and ii*), but dramatically reduced the expression of Class II (*Bmp7*, *Tfrc*, *Crym1*, *Uncx4*, *Etv5* and *Slc12a*) target genes within progenitors (*Figure 4D* and *Figure 4—figure supplement 1C*). Interestingly, *Tfrc* was absent from mesenchymal cells, but was maintained to be expressed in the ureteric epithelium suggesting that progenitor-specific R-spondin

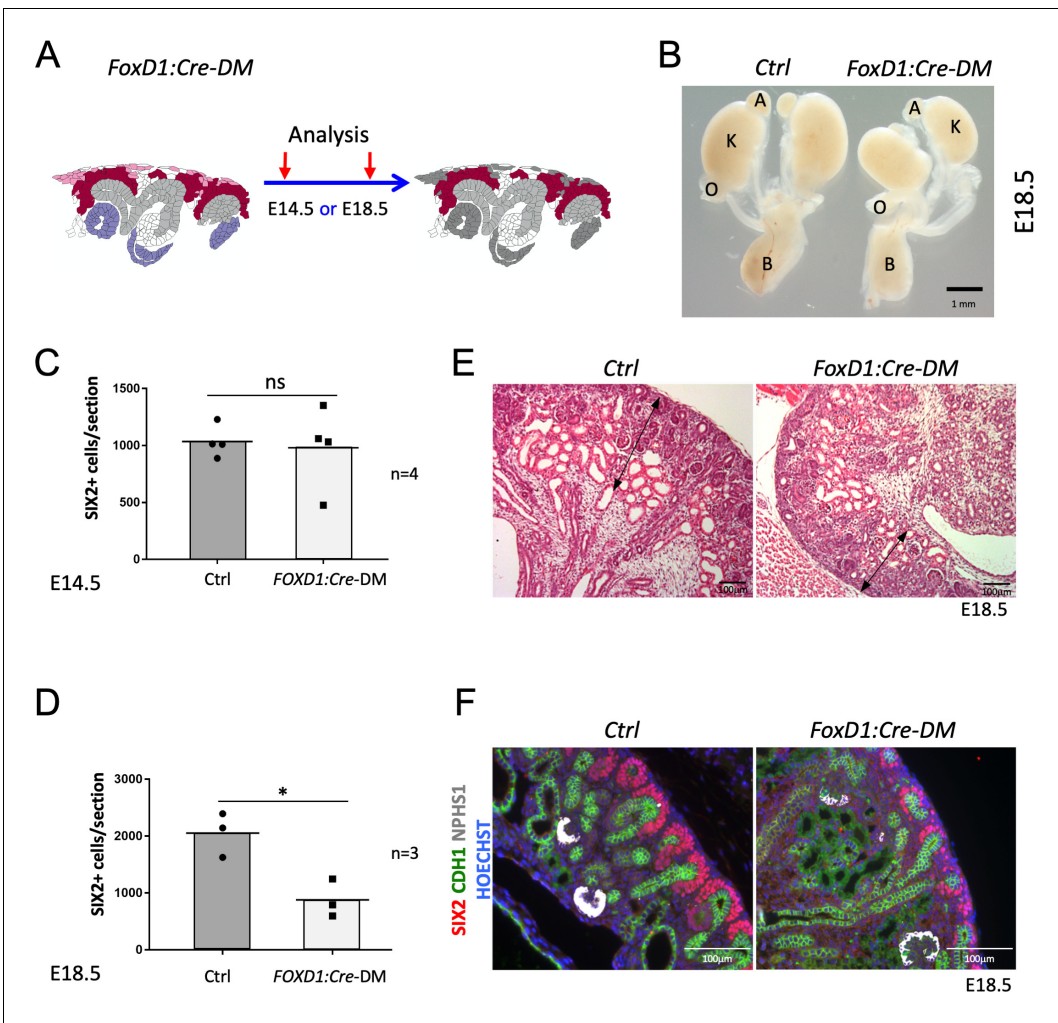

**Figure 3.** Stromal RSPO3 maintains the pool of renal progenitors at late stages of kidney development. (**A**) Schematic outline of the strategy used for stromal-specific deletion of *Rspo3* in the absence of *Rspo1*. (**B**) Macroscopic view of urogenital systems at E18.5 reveals smaller kidneys in *Foxd1:Cre-DM* mutants when compared to control littermates. (**C**) Quantification of SIX2[+] progenitors reveals no significant difference in the number of progenitors between mutant and control kidneys at E14.5 (n = 4 embryos for each genotype, one litter), see *Figure 3—source data 1* (**D**) but a more than 50% decrease by E18.5 (n = 3 embryos for each genotype, two litters), see *Figure 3—source data 1*. (**E**) H and E staining of kidney sections at E18.5 shows a reduction of the nephrogenic zone (double arrowed black lines). (**F**) IF analysis using anti-CDH1 (green) anti-SIX2 (red), and anti-NPHS1 (marks podocytes in white) antibody reveals a loss of progenitors. Nuclei were counterstained with Hoechst (blue). A: adrenal gland, B: bladder, K: kidney, O: ovary. Columns are means ± SEM with p<0.05 (*), p<0.01 (**), p<0.001 (***), p<0.0001 (****). Foxd1:Cre-DM stands for (Foxd1:Cre, Rspo1[-/-], Rspo3[fl/fl]) double mutant, Ctrl: Control.

The online version of this article includes the following source data for figure 3:

**Source data 1.** Progenitors quantification in *Foxd1:Cre-DM* at E14.5 and E18.5.

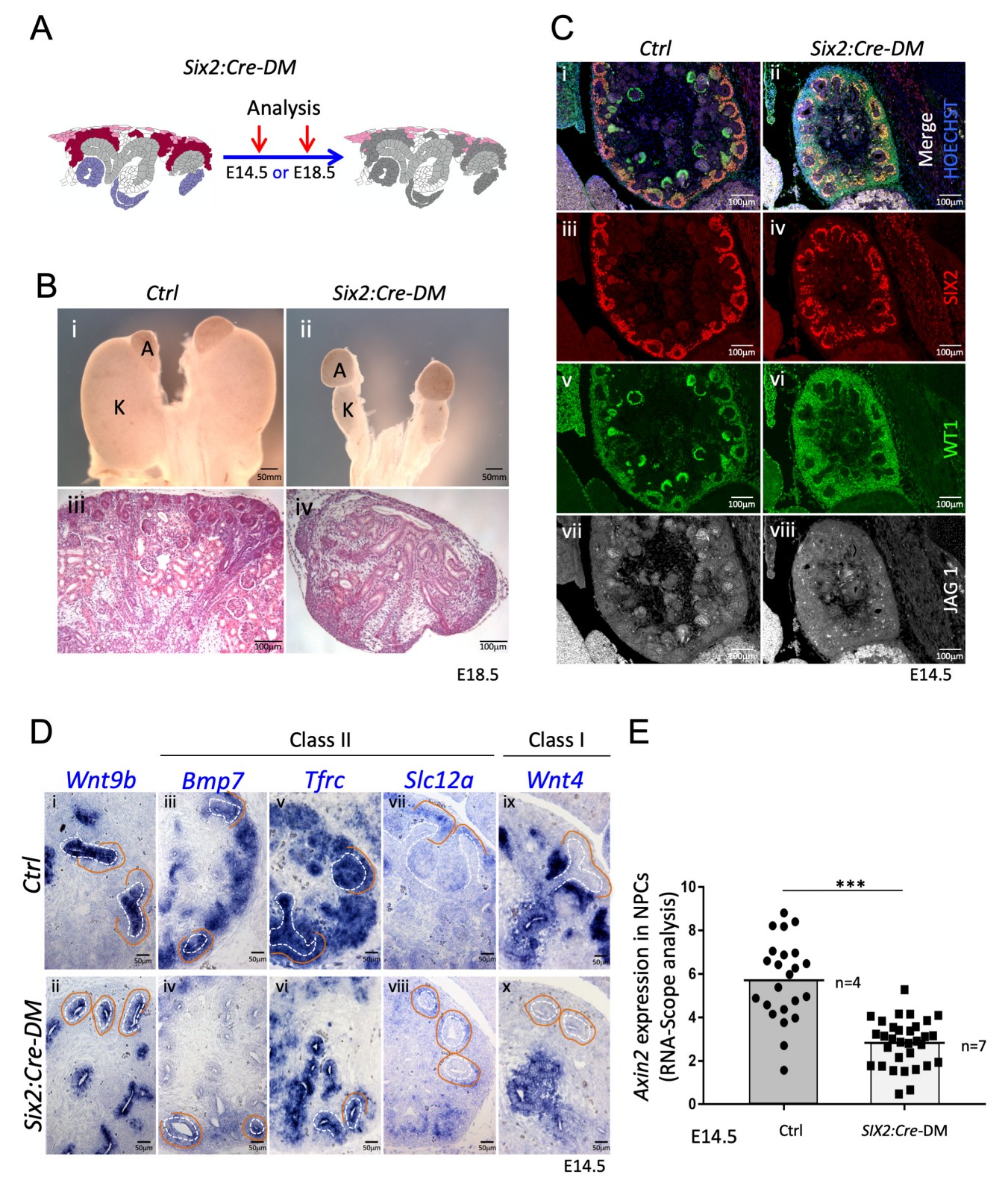

**Figure 4.** Absence of R-spondins from progenitors causes lack of MET and downregulation of β-catenin target genes. (**A**) Schematic outline of the strategy used for progenitor-specific deletion of *Rspo3. Six2:Cre-DM stands for (Six2:Cre, Rspo1^{-/-}, Rspo3^{fl/fl}).* (**B**) Macroscopic view reveals smaller kidneys (K) in mutant E18.5 embryos. H and E staining reveals a complete absence of glomeruli (compare **iii** and **iv**). (**C**) Immunolabelling for SIX2 (red), WT1 (green) and JAG1 (white) revealed a mild reduction of progenitors and confirmed the lack of nephrons on the molecular level (**D**). In situ
*Figure 4 continued on next page*

Figure 4 continued

hybridization performed on E14.5 embryos revealed persistence of *Wnt9b* expression, but dramatic reduction of class I (*Wnt4*) and class II (*Bmp7, Tfrc, Slc12a*) β-catenin target genes in the nephrogenic lineage. Dotted white lines highlight the ureter and orange lines outline the CM compartment. (E) Quantification with RNA-Scope technology combined with Halo software analysis shows a reduction of *Axin2* expression by 51% in the nephrogenic zone of mutant kidneys compared to control (p<0.0001). (n = 4 embryos isolated from four litters for control genotype, and n = 7 embryos isolated from six litters for mutant genotype). Each dot or square represents the total RNAScope signal detected per nephrogenic area normalised to the total number of cells present in this field. See *Figure 4—source data 1*. Columns are means ± SEM with p<0.05 (*), p<0.01 (**), p<0.001 (***), p<0.0001 (****).

The online version of this article includes the following source data and figure supplement(s) for figure 4:

Source data 1. Source data for *Figure 4E*: Quantification of *Axin2* RNA-Scope signal.
Figure supplement 1. Analysis of progenitor-specific deletion reveals a requirement of R-Spondins for MET.

expression is dispensable for its activation in the collecting duct system. The class I target *Wnt4*, a key regulator of MET (*Stark et al., 1994*), was also undetectable within the nephrogenic zone, but remained expressed in medullary stromal cells (*Figure 4D* ix and x). Consistent with a requirement of R-spondins for β-catenin activation, the direct downstream target *Axin2* was 51% down regulated in the nephrogenic zone of mutant kidneys (p<0.0001, *Figure 4E*; *Figure 4—source data 1* and *Figure 4—figure supplement 1D*).

*Six2:Cre-DM* kidneys showed a small reduction of nephron progenitors (*Figure 4C*). To exclude the possibility that the lower number of progenitors interfered with the MET process, we employed the *Wt1:CreER^{T2}* strain (*Figure 5A*), which in response to Tamoxifen activation, induced deletion primarily within nephron progenitors (*Figure 1—figure supplement 2* and *Figure 5—figure supplement 1*). *Wt1:CreER^{T2}-DM* embryos kidneys contained large numbers of SIX2^+ progenitors (*Figure 5Bi and ii*) that failed to differentiate. Indeed, expression of LEF1, a direct target and interaction partner of β-catenin that is also a marker of committed pretubular aggregates, was virtually absent from the nephrogenic region of *Rspo1/3* double mutant mice (*Figure 5Biii and iv*). *Six2:Cre* and *Wt1:CreER^{T2}* driven deletion thus showed a very similar phenotype. The higher number of progenitors in *WT1:CreER^{T2}*-induced mutants is likely due to less efficient deletion of *Rspo3* and, as a consequence, persistence of slightly higher levels of R-spondin activity.

Isolated NPCs can be induced to undergo MET when cultured under high-density conditions and stimulated with chiron (CHIR99021), an inducer of canonical β-catenin signalling. To test whether R-spondins can replace chemical inducers under such in vitro conditions, we treated freshly isolated NPCs with recombinant WNT3A, RSPO3 or a combination of both (*Figure 5C*; *Figure 5—source data 1*). Gene expression analysis after 48 hr revealed a 2.3-fold and 4.6-fold increase for the β-catenin targets *Axin2* and *Wnt4*, respectively, after combined treatment (*Figure 5D*). Interestingly, treatment with recombinant WNT3A alone did not have an effect suggesting that WNT/β-catenin signalling is actively suppressed in this cell type and crucially requires R-spondin action to permit activation of this pathway (*Figure 5D*).

Commitment of renal progenitors to nephron differentiation involves canonical BMP7 signalling, which allows cells to fully respond to the WNT/β−catenin pathway and undergo MET (*Brown et al., 2013*). In situ hybridization indicated a dramatic downregulation of *Bmp7* in *Six2:Cre-DM* kidneys (*Figure 4D*). Canonical BMP signalling induces phosphorylation of SMAD1/5 and we therefore evaluated the phosphorylation status of this signal transducer. Whereas progenitors in control kidneys that engaged in MET were positive for pSMAD1/5 (*Figure 5E*), the nephrogenic zone in *Six2:Cre-DM* kidneys showed a 88% reduction of staining (p<0.0001; *Figure 5F*; *Figure 5—source data 1*). pSMAD staining in medullary stroma was not affected by this deletion. Taken together these data demonstrate that R-spondins are required for *Bmp7* activation, which in turn permits priming of renal progenitors for MET.

## LGRs are dispensable for MET

R-spondins act by associating and internalising the ubiquitin E3 ligases ZNRF3/RNF43. Expression analysis using RNA-Scope demonstrated RNF43 to be specifically expressed within the branching ureter, whereas ZNRF3 showed a more widespread expression pattern, covering also the nephrogenic niche (*Figure 6—figure supplement 1*). R-spondin interaction with ZNRF3/RNF43 is believed to be primarily mediated via binding to their cognate receptors LGR4-6 and mutations in *Lgr4* or

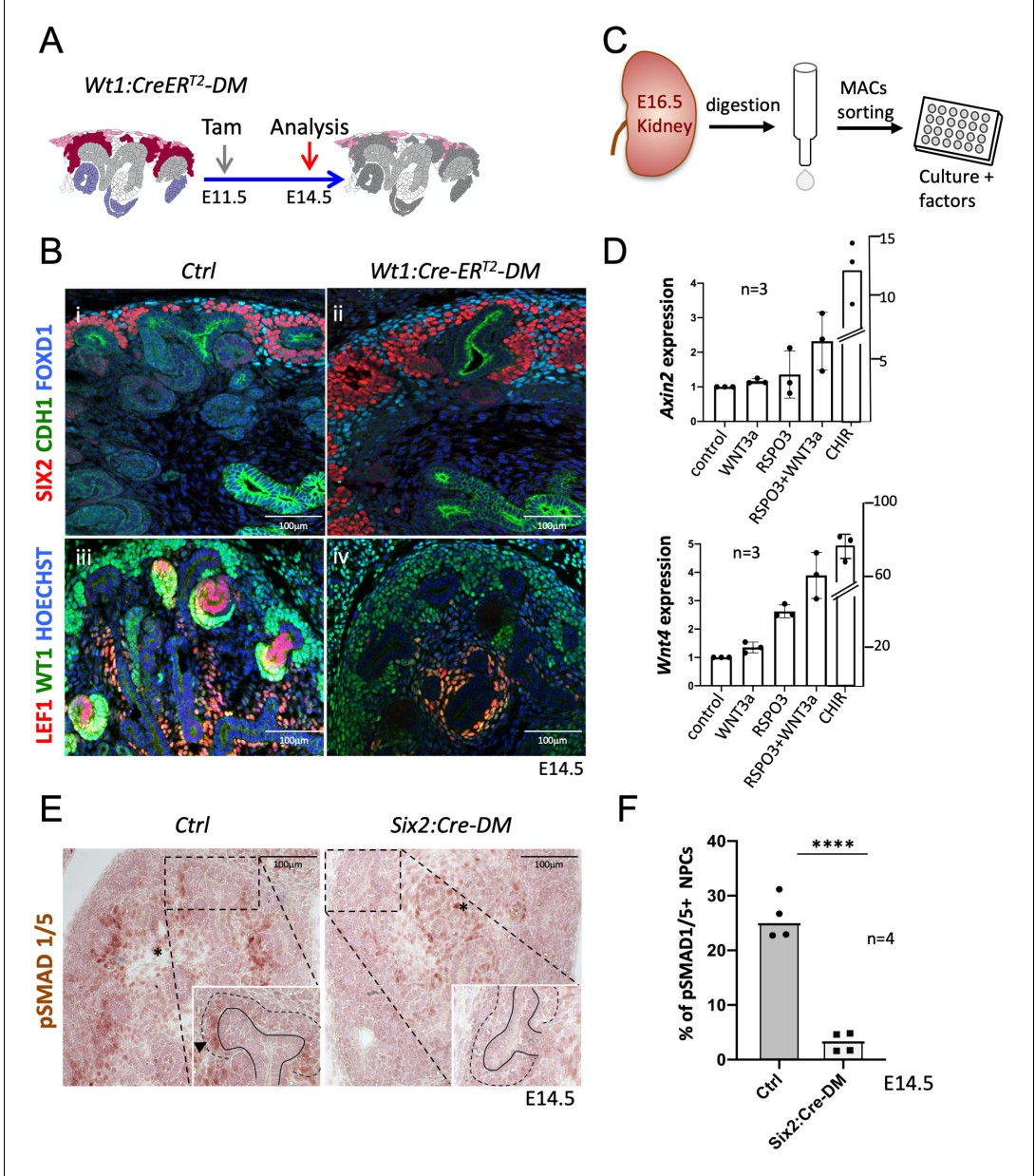

**Figure 5.** R-spondins are required for MET induction. (**A**) Schematic outline of the strategy used for *Wt1:CreER^T2* induced deletion of *Rspo3. Wt1: CreER^T2-DM stands for (Wt1:CreER^T2, Rspo1^-/-, Rspo3^fl/fl)*. (**B**) **i** and **ii**) Immunolabelling revealed lack of nephrogenesis upon *Wt1:CreER^T2* induced deletion of *R-spondins*, despite the persistence of large numbers of SIX2^+ (red) nephron progenitors and FOXD1^+ (blue) stroma progenitors (CDH1 = green). **iii** and **iv**) Staining for LEF1 (red) and WT1 (green) confirmed the lack of nephrogenesis (**C**) Schematic representation of the methodology followed to isolate and grow kidney progenitors in vitro. (**D**) Comparison of *Axin2* or *Wnt4* gene expression levels in nephron progenitors treated by recombinant protein WNT3a (50 ng/ml) or RSPO (200 ng/ml) alone, WNT3a and RSPO3 in combination or CHIR (3μM) alone as an internal positive control. Experiments were performed as triplicates (n = 3). Compared to control, addition of RSPO3 + WNT3A leads to a 2.3 fold and 4.6 fold increase of *Axin2* and *Wnt4* expression respectively, see *Figure 5—source data 1*. (**E**). Immunohistochemical analysis for pSMAD1/5 demonstrated the lack of nephron progenitor priming (black arrowhead). Note the persistence of pSMAD staining in medullary stroma (black asterisk). In the inset, black lines outline the ureter and dotted lines the CM compartment. (**F**) Quantification of progenitors that are pSMAD1/5^+ reveals a highly significant reduction of SMAD1/5 phosphorylation in mutant compared to control kidneys at E14.5 (n = 4 embryos for each genotype, 3 and 4 litters for control and mutant respectively), see *Figure 5—source data 1*.

The online version of this article includes the following source data and figure supplement(s) for figure 5:

**Source data 1.** Quantification of *Wnt4* and *Axin2* expression in progenitors cultured in vitro and quantification of pSMAD1/5^+ progenitors in *Six2:Cre-DM* kidneys.

**Figure supplement 1.** *Wt1:CreER^T2* induced deletion leads to reduced *Rspo3* expression.

*Lgr4/5* have been shown to have mild defects in kidney development (*Kato et al., 2006*; *Dang et al., 2014*; *Kinzel et al., 2014*). To determine to what extent RSPO1/3 activity in kidney development depends on the LGR receptor family we mapped their expression using RNA-Scope analysis. *Lgr4* showed the broadest expression pattern with mRNA detected in virtually all cell types of the developing kidney (*Figure 6A*). Particularly, strong staining was found in the distal part of comma shaped bodies that extended into the intermediate segment at the S-shaped body stage. In addition, we detected strong staining in medullary stromal cells that line the developing ureter. *Lgr5* mRNA was virtually absent from the CM, but could be found within ureteric tips, a proportion of medullary stroma cells and the distal segment of S-shaped bodies, as previously reported using a lacZ reporter strain (*Barker et al., 2012*). *Lgr6* mRNA showed the most restricted expression pattern and could only be detected in PTAs of forming nephrons (*Figure 6A*).

Based on these findings we hypothesized that LGR4 was likely to be the main receptor mediating RSPO function in renal progenitors. To test this hypothesis, we took advantage of a mouse strain carrying an *Lgr4* null allele that was previously generated by our laboratory (*Da Silva et al., 2018*). Quantification of SIX2-positive NPCs at E14.5 indicated a reduction by 37% (p=0.0006) in *Lgr4*$^{-/-}$ mutants compared to control kidneys (*Figure 6B,C*; *Figure 6—source data 1*), a cell loss that was considerably inferior to that observed in *R-spondin* mutants. Moreover, MET appeared to be unaffected in LGR4 mutants as exemplified by the activation of LEF1 in PTAs and the presence of WT1 positive cells in proximal proportion of forming nephrons (*Figure 6D*). To test whether LGR5 and/or LGR6 may compensate for the lack of LGR4, we next analyzed wholebody *Lgr4/5/6* triple knockout kidneys at E14.5, the latest developmental stage where these embryos can be collected. The absence of LGR4/5/6 had little effect on early renal development and progenitors were able to undergo MET and form WT1+ glomeruli (*Figure 6E*). We conclude that while LGR4 appears to mediate some R-spondin activity within the nephrogenic niche, the majority of signalling occurs in an LGR-independent manner.

## Discussion

Controlled proliferation and differentiation of renal progenitors is essential for the establishment of sufficient numbers of nephrons that ensure healthy kidney function throughout life. Our analysis has established R-spondins as novel key regulators that ensure the maintenance of a healthy pool of nephron progenitors throughout kidney development and permit MET during nephrogenesis (*Figure 6F*).

Consistent with their partially overlapping expression pattern, *R-spondins* appear to act in a functionally redundant manner in kidney development and single gene deletion of *Rspo1* or *Rspo3* had only mild effects on progenitor numbers (*Figure 1—figure supplement 3*). However, whereas *Rspo1* is present in all SIX2$^+$ cells, *Rspo3* expression is restricted to a subpopulation that - based on their location at the outermost cortex - appear to represent 'ground-state' progenitors. Interestingly, by E18.5 *Rspo3* expression is lost from the nephron progenitor compartment, which is likely to reflect the progressive aging/differentiation of these cells. Indeed, age-dependent changes of nephron progenitors that affect their proliferation capacity and lifespan have been described, previously (*Chen et al., 2015*).

At late stages of embryogenesis, *Rspo3* expression becomes almost exclusively restricted to stromal progenitors, which is consistent with recent single sequencing data at E18.5 that highlighted *Rspo3* as a marker for the stromal compartment (*Combes et al., 2019*). Stromal expression appears to be important for nephron progenitor maintenance, as *Rspo3* deletion in this compartment (*Foxd1:Cre*) resulted in progenitor loss at later stages of development. Renal progenitor cells are thus sandwiched between stromal cells releasing RSPO3 and the WNT9b producing ureteric bud that ensures their survival at later stages of development.

In previous studies, R-spondins, and in particular *Rspo3*, have been shown to enhance both canonical and non-canonical WNT signalling (*Kazanskaya et al., 2004*; *Scholz et al., 2016*; *Ohkawara et al., 2011*). Our analysis demonstrates that upon R-spondin deletion direct downstream targets of β–catenin, such as *Tafa5* (*Karner et al., 2011*) and *Axin2* (*Lustig et al., 2002*) are reduced in nephron progenitors indicating that in kidney development R-spondins activate canonical WNT signalling. Indeed, the observed phenotypes largely mimic those seen upon loss of β–catenin activity (*Karner et al., 2011*; *Carroll et al., 2005*; *Park et al., 2012* and this study). These findings are

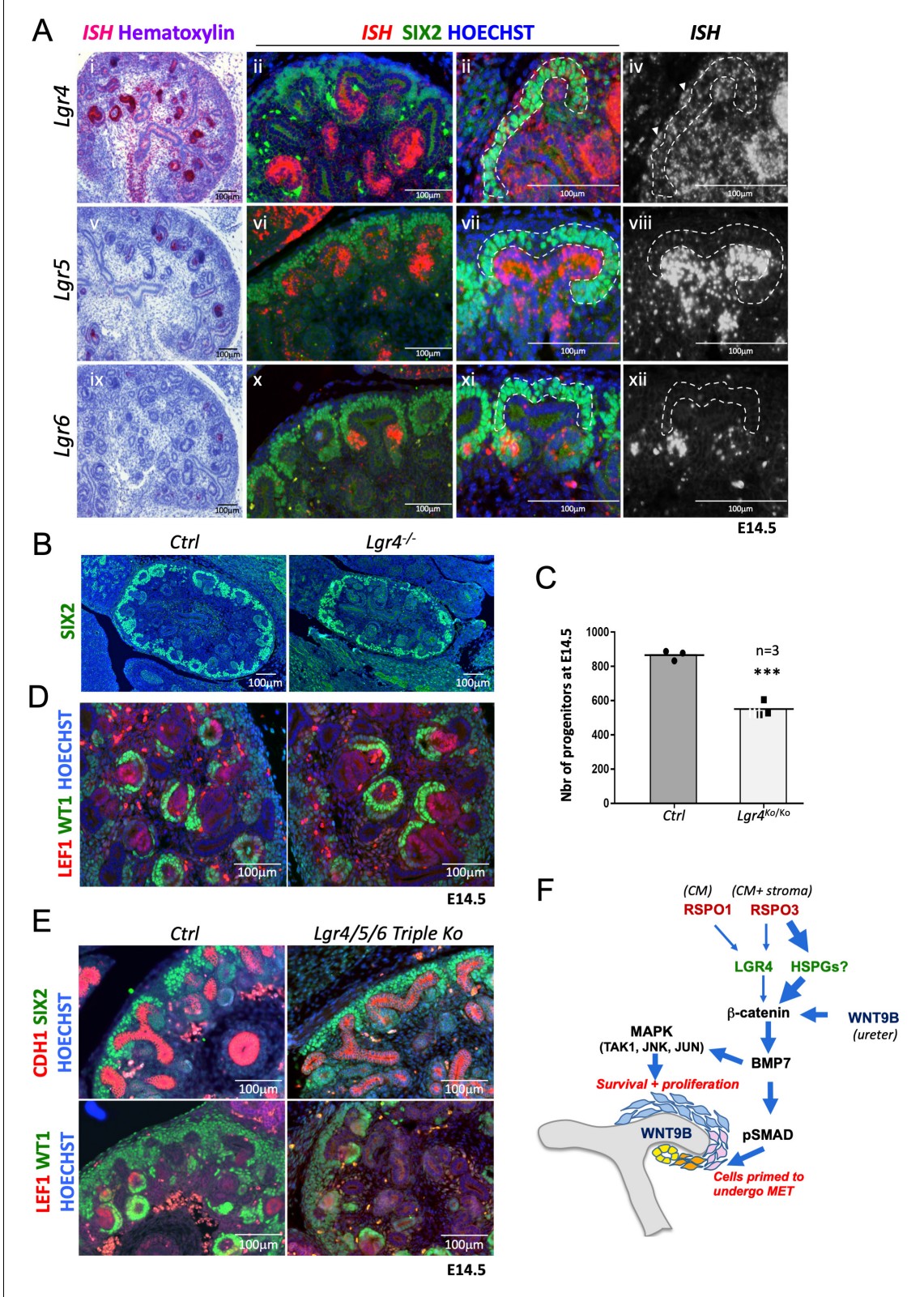

**Figure 6.** R-spondins can function in an LGR-independent manner during kidney development. (**A**) i-iiii) *RNAScope* analysis (red) revealed low levels of *Lgr4* expression throughout the developing kidney with strong signal within the distal portion of the forming nephron and weak activation in the stromal cells (white arrowhead). **v-viii**) *Lgr5* expression was found within the ureteric tip and distal segment of S-shaped bodies. **ix-xii**) *Lgr6* expression was restricted to PTAs of newly forming nephrons. (SIX2 = green, Hoechst = blue). The Cap Mesenchyme compartment is outlined by dotted white

*Figure 6 continued on next page*

*Figure 6 continued*

lines. (B) Immunofluorescence analysis of *Lgr4* knockout and control samples performed on E14.5 kidney sections with SIX2 antibodies reveals a reduction of nephron progenitors. (C) Quantification of SIX2⁺ progenitors from (B) (for each genotype n = 3 embryos collected from two litters,) Every black dot or square represents the total number of SIX2+ progenitors located in the CM counted in one control or mutant of an entire kidney field, see *Figure 6—source data 1*. (D) *Lgr4* negative progenitors undergo MET as revealed by WT1 staining (high WT1 expression is found in the proximal part of Comma and S-shaped bodies, as well as podocytes). (E) Immunofluorescent analysis in wholebody *Lgr4/5/6* mutants demonstrates persistence of progenitors and MET despite the absence of all three cognate R-spondin receptors. (F) Model for the molecular cascade regulated by R-spondins during nephrogenesis.

The online version of this article includes the following source data and figure supplement(s) for figure 6:

**Source data 1.** Source data for *Figure 6C*: Quantification of progenitor cells (SIX2+) in controls and *Lgr4* mutants at E14.5.
**Figure supplement 1.** *RNF43* and *ZNRF3* expression pattern in developing kidneys.

consistent with recent in vitro data that described *Rspo1* to enhance canonical signalling upon treatment of a mesonephric cell line (M15) with WNT9b (*Dickinson et al., 2019*). However, at present we can not exclude that R-spondins may also modulate non-canonical WNT signalling, which might affect cell adhesion and migration of NPCs. Live imaging experiments in *Six2-Cre:DM* will be required to address this hypothesis.

The substantially different phenotypes observed in *CAGG:CreER*ᵀᴹ (or *Foxd1:Cre*) driven deletion of R-spondins, which impacted progenitor survival, and *Wt1:CreER*ᵀ² (or *Six2:Cre*) driven excision that interfered with MET, may at a first glance seem surprising. However, the ubiquitously expressed *CAGG:CreER*ᵀᴹ strain completely abolished *R-spondin* expression in the kidney and thus effectively blocked β–catenin signalling within progenitors. Since β–catenin signalling is essential for proliferation and survival, progenitor cell numbers rapidly declined in these mutants. By contrast, *Six2:Cre* and *Wt1:CreER*ᵀ²-induced deletions did not interfere with stromal *Rspo3* expression, which bestowed sufficient β-catenin signalling on progenitors to permit survival. This hypothesis is compatible with a model in which low and high levels of β–catenin signalling regulates Class II and Class I target genes, respectively (*Ramalingam et al., 2018*). However, the concept of lower β–catenin signalling in uncommitted progenitors seems contradictory considering that they are exposed to higher levels of R-spondins (RSPO1+RSPO3), when compared to cells that engage in MET (only RSPO1). This dilemma can be resolved when introducing an additional 'switch' that permits the activation of class I targets (e.g. *Wnt4/Lef1*) once progenitors leave the cortical niche. Evidence from knockout studies suggests this switch to involve *Bmp7*-dependent activation of pSMAD signalling at the boundary between the niche and committed cells (*Brown et al., 2013*). Since progenitors in *Six2:Cre-DM/Wt1:CreER*ᵀ²-DM mice have significantly decreased levels of *Bmp7* (a class II target of β-catenin [*Park et al., 2012*]), progenitor cells fail to activate pSMAD signalling, lack expression of class I targets and as a consequence do not epithelialize.

We have also addressed the potential role of LGR4/5/6, the cognate receptors of R-spondins, in conveying their action. *Lgr4* single mutants have been described previously to show dilated collecting ducts and reduced kidney size (*Kato et al., 2006*) that could be traced back to a reduction in renal progenitors (*Dang et al., 2014*). Our own analysis with an independent *Lgr4* knockout allele confirms this phenotype, although duct dilation was only seen at late stages of kidney development (*Da Silva et al., 2018* and data not shown). In R-spondin mutants we did not observe dilated collecting ducts, which might be explained by persistent expression of *Rspo3* in medullary stroma using the tissue-specific deletion in stromal or nephron progenitors. *Lgr5* was found to be virtually absent from uninduced progenitors, but was strongly expressed within the ureteric tip and the distal part of the developing nephron, two sites of strong β-catenin activity. To our knowledge, no renal abnormalities have been associated with *Lgr5* mutations and the previously published *Lgr4/Lgr5* double mutants show a kidney phenotype comparable to the *Lgr4* single mutant (*Kinzel et al., 2014*). Our RNAScope analysis revealed *Lgr6* expression to be highly restricted to the pretubular aggregates and renal vesicle. Surprisingly, *Lgr4/5/6* triple knockouts are associated with only a mild loss of renal progenitors that are capable to undergo MET, a phenotype that does not phenocopy the dramatic loss of renal progenitors and their inability to undergo MET correlated with the absence of *Rspo1* and *Rspo3*. Due to the embryonic death around E14.5, later stages could not be analysed. Taken together these data indicate that *Lgr4*, but not *Lgr5/6*, contribute to R-spondin signalling in nephron progenitor cells. However, the mild phenotype observed in single *Lgr4* mutants and the triple *Lgr*

knockout when compared to *Rspo1/3* mutants, suggests that R-spondins are likely to act primarily in an LGR-independent manner in NPCs. A similar LGR-independent action has recently been described for *Rspo2* in limb and lung development (*Szenker-Ravi et al., 2018*). Interestingly, an alternative mode of action for RSPO2 and RSPO3 has recently emerged that potentiates the WNT/β−catenin pathway through an interaction with heparan sulfate proteoglycans (HSPG) (*Lebensohn and Rohatgi, 2018*). In *Xenopus*, RSPO3 has been shown to interact with Syndecan4 to induce non-canonical (PCP) signalling (*Ohkawara et al., 2011*) and Syndecan1 has been suggested to present WNT and R-spondins in multiple myeloma (*Ren et al., 2018*). Further analysis will be needed to identify the proteins responsible for mediating LGR-independent R-spondin action in the kidney.

The number of nephrons varies dramatically in the human population with a count of 250,000 and 2.5 million glomeruli per kidney being considered as normal. However, lower nephron numbers have been directly linked with hypertension and a higher risk of developing renal diseases (*Bertram et al., 2011*). Our study has identified R-spondins as regulators that maintain the nephrogenic niche during development and we can speculate that changes in expression levels or protein function in these genes may be associated with renal disorders. Interestingly, the *RSPO3* locus has been identified in two studies to be linked with renal diseases including abnormal blood urea nitrogen (BUN), a hallmark for glomerular filtration dysfunction (*Okada et al., 2012*; *Osman et al., 2018*). Given these studies and our findings in mice, it will be of interest to further investigate a potential role of R-spondin variants in the predisposition to renal disease.

## Materials and methods

### Mice

The experiments described in this paper were carried out in compliance with the French and international animal welfare laws, guidelines and policies and were approved by the local ethics committee (PEA N°: NCE-2014–207 and PEA N°: 2018060516474844 (V2)). Genetically modified mice used in this study have been described previously: *Rspo3^{flox}* (*Rocha et al., 2015*), *Rspo1^-* (*Chassot et al., 2008*), *Ctnnb^{flox}* (*Brault et al., 2001*), *Lgr4^-* (*Da Silva et al., 2017*), *Lgr4/5/6^-*(*Szenker-Ravi et al., 2018*), *CAGG:CreER^{TM}* (*Hayashi and McMahon, 2002*), *Wt1:CreER^{T2}* (*Zhou et al., 2008*), *Six2:Cre* (*Kobayashi et al., 2008*) and *Foxd1:Cre* (*Kobayashi et al., 2014*). Primers for genotyping can be found in *Table 1*. For inducible Cre lines (*CAGG:CreER^{TM}* and *Wt1:CreER^{T2}*), Cre activation was achieved by gavage of pregnant females with an extemporarily prepared single dose of 200 mg tamoxifen dissolved in corn oil per kg of body weight. For proliferation assays, BrdU dissolved in 0.9% NaCl was administered to pregnant dams 2 hr before sacrificing via intraperitoneal (IP) injection at a dose of 50 mg/kg. Embryos were analyzed at various time-points and genders were not considered.

### Embryo collection

Embryos were collected from time-mated females considering the date when the vaginal plug was observed as embryonic day 0.5 (E0.5). Embryos from E11.5 to E18.5 were fixed with 4% paraformaldehyde (PFA) in phosphate buffer saline (PBS) overnight at 4°C or at room temperature, if used for RNA-Scope analysis. The day after, embryos were washed in PBS, dehydrated through an Automatic tissue processor (Leica TP1020) and embedded in paraffin.

### In situ hybridization

Tissues were fixed overnight in 4% paraformaldehyde, progressively dehydrated and embedded in paraffin. 7-μm-thick sections were cut then rehydrated and hybridization was performed as described in *Lescher et al., 1998* using an *InsituPro VSi* robot from Intavis Bioanalytical Instruments. Digoxygenin-labelled antisense RNA probes were synthesized from plasmids obtained from different sources: *Rspo1*, *Rspo3*, *Wnt4*, *Wnt9b* (Gift from McMahon's laboratory), *Bmp7*, *Etv5* and *Crym* (Gift from T. Carroll's laboratory), *Uncx4* (Gift from A. Reginensi, in J. Wrana's laboratory), *Slc12a*, *Tfrc* and *Tafa5* DNA sequences were subcloned in pCRII Topo vector. Hybridized DIG-RNA probes were detected with alkaline phosphatase-coupled anti-digoxigenin antibody. After washing, the chromogenic reaction was performed with BM-purple substrate for several days at room temperature.

**Table 1.** List of primer pairs used for qPCR or genotyping analysis in this study.

| Name | Forward | Reverse | Usage |
|------|---------|---------|-------|
| Axin2 | GCAAGTCCAAGCCCCATA | CGGCTGACTCGTTCTCCT | qPCR |
| Rspo2 | GCCAGGCAAAAGACACAATACCA | CAATCAGGCTTCCGCCTCTTCTTC | qPCR |
| Rspo3 | TCAAAGGGAGAGCGAGGA | CAGGAGGAGGAGCTTGTTTCC | qPCR |
| Rspo4 | GCTGTTTCAGGCCAGGGACTTC | TTGTGTATGCAGGGGACTCCA | qPCR |
| Sdha | TGTTCAGTTCCACCCCACA | TCTCCACGACACCCTTCTG | pPCR |
| Wnt4 | ACTGGACTCCCTCCCTGTCT | TGCCCTTGTCACTGCAAA | qPCR |
| Ctnnb1 | AAGGTAGAGTGATGAAAGTTGTT | CACCATGTCCTCTGTCTATTC | Genotyping |
| Cre | CCCACCGTCAGGTACGTGAGATAT | CGCGGTCTGGCAGGTAAAAACTAT | Genotyping |
| Foxd1 Wt | CTCCTCCGTGTCCTCGTC | TCTGGTCCAAGAATCCGAAG | Genotyping |
| Foxd1 Cre | GGGAGGATTGGGAAGACAAT | TCTGGTCCAAGAATCCGAAG | Genotyping |
| Lgr4 Wt and Ko | TGTGTTTTGGCTTGCTTGAC | AGTCTGCTCCCCTACCACT | Genotyping |
| Lgr4 Wt | TGCAACCCTAGAAGGGAAAA | CTCACAGGTGCTTGGGTGAAG | Genotyping |
| Lgr4 Ko | GCCTGCATTACCGGTCGATGCAACGA | CTCACAGGTGCTTGGGTGAAG | Genotyping |
| Lgr5 Wt | ACATGCTCCTGTCCTTGCT | GTAGGAGGTGAAGACGCTGA | Genotyping |
| Lgr5 Ko | CACTGCATTCTAGTTGTGG | CGGTGCCCGCAGGCGAG | Genotyping |
| Lgr6 Wt | CGCTCGCCCGTCTGAGC | GCGTCCAGGGTCCGCAGGG | Genotyping |
| Lgr6 Ko | CGCTCGCCCGTCTGAGC | CCTGGACGTAGCCTTCGGGC | Genotyping |
| Rspo1Ko | ATCCAGGGGTCCCTCTTGATC | AATATCGCGGCTCATTCGAGG | Genotyping |
| Rspo1Wt | ATCCAGGGGTCCCTCTTGATC | TTGAGGCAACCGTTGACTTC | Genotyping |
| Rspo3flox | CTTCAACTTGAAGGTGCTTTACC | CCAGGAATGTACAACAGGATCCTCTC | Genotyping |

*RNAscope* analysis (Advanced Cell Diagnostics; *Lgr4*, *Lgr5*, Lgr6, *Rspo1*, *Rspo2*, *Rspo3*, *Axin2*) was performed according to the manufacturer's instructions using the chromogenic Fast Red dye that can be visualized using light or fluorescence microscopy. Alternatively, after in situ hybridization, sections were blocked in a PBS solution containing 3% BSA, 10% Normal Donkey Serum and 0.1% Tween, then primary antibodies were added at concentrations reported in *Supplementary file 1* and incubated overnight at 4°C. The following day, after three washes in PBS, secondary antibodies were diluted 1/500 in PBS and applied on sections for 1H at room temperature. After three washes in PBS, sections were mounted in a 50% glycerol (in PBS) medium.

## Immunofluorescence and histological analysis

For immunofluorescence experiments, tissues were fixed overnight in 4% paraformaldehyde, progressively dehydrated and embedded in paraffin. 5 μm thick sections were rehydrated, boiled in a pressure cooker for 2 min with Antigen Unmasking Solution and blocked in PBS solution containing 10% normal donkey serum, 0.1% tween and 3% BSA. All antibodies were applied overnight at 4°C at the concentrations listed in the *Supplementary file 1*. Secondary antibodies were diluted 1:500 and applied at room temperature for 1 hr. For histological analysis, 5-μm-thick sections were stained with hematoxylin and eosin according to standard procedures.

## Quantitative RT-PCR analysis

RNA was extracted from embryonic samples or cultured cells using RNeasy Mini- or Micro-kit, following the manufacturer's instructions. Reverse transcription was performed using M-MLV reverse transcriptase in combination with Random Hexamers. The cDNA synthesised was used as a template for qPCR reaction performed using the Light Cycler SYBR Green I Master Kit. Expression levels were normalized for *Sdha*. Primers are described in *Table 1*.

## Quantification of proliferating renal progenitors

A combination of anti-BrdU and anti-SIX2 antibodies was used to detect proliferating renal progenitors. Quantification was performed on four embryos (n = 4) isolated either from 2 (for *CAGG: CreER^{TM}-DM*) or 3 (for Ctrl) different litters. For each embryo between 10 and 40 kidney sections were analysed that were collected throughout the entire organ. Pictures were taken with a Zeiss Apotome two upright microscope and every kidney field on the picture was analysed with Fiji software. Only SIX2+ cells located in the cap mesenchyme were considered for quantification. The percentage of proliferating progenitors was obtained after scoring for every embryo the average of (BrdU/SIX2) double positive nuclei divided by the total number of SIX2+ cells in the CM. Each value was reported on the graph as a single symbol (black dot for Ctrl and black square for *CAGG: CreER^{TM}-DM*).

## Quantification of apoptotic renal progenitors

Apoptotic cells were labelled with TUNEL kit (Roche) and renal progenitors identified using anti-SIX2 antibodies. Quantification was performed on three embryos (n = 3) isolated from two different litters. For each embryo, between 6 and 15 sections, going through the kidney centre (pelvic region) were analysed. Pictures were taken for every kidney with a Zeiss Apotome two upright microscope. In every kidney field, only SIX2+ cells located in the cap mesenchyme were considered and quantified with Fiji software. The percentage of apoptotic progenitors was obtained after scoring for every embryo the average of (TU/SIX2) double positive nuclei divided by the total number of SIX2+ cells in the CM. Each value was reported on the graph as a single symbol (black dot for Ctrl and black square for *CAGG:CreER^{TM}-DM*).

## Quantification of renal progenitors

SIX2$^+$ progenitors located in the cap mesenchyme were counted in three sagittal sections (20 µm apart to each other) of kidney samples through the renal pelvis and the average number obtained for every embryo was reported on the graph.

## Quantification of Axin2 level of expression from RNAScope analysis

Three to eight sagittal sections of kidney samples were cut through the renal pelvis (20 µm apart from each other), and subjected to in situ RNA hybridisation (RNAScope technology) using an *Axin2* anti-sense probe. Sections were mounted in Vectashield antifade mounting medium with DAPI and imaged with a Slide scanner Vectra Polaris. RNAScope signal was detected with filter 555 and nuclei with filter 350. Images were analysed with Halo software (Indica labs) using FISH-IF module that was based on cell segmentation and ISH signal per cell. The value obtained for every kidney section analysed is reported on the graph by either a point (for control) or a square (for mutant). Each symbol corresponds to the total RNAScope signal detected per selected kidney field normalised to the total number of cells present in this area.

## Quantification of progenitors that are pSMAD1/5 positives

Immunostaining with anti-pSMAD1/5 antibodies was performed on 3 to 6 sagittal sections (20 µm apart to each other) of kidney samples, and signal was revealed using the Vector Red alkaline phosphatase kit. For every section, progenitors located in CM that were labelled with anti-PSMAD1/5 antibodies were counted and normalised to the total number of progenitors that were in this CM. Every average number obtained for every embryo was reported on the graph as a dot (for control) or a square (for mutant) samples.

## Statistical analyses

Data are shown as mean (± s.e.m). Analyses were performed according to the two-tailed unpaired Student's *t*-test, *p<0.05, **p<0.01, ***p<0.001, ****p<0.0001. The letter 'n' refers to the number of individual samples.

## Nephron progenitor isolation and culture

Kidneys were dissected from E16.5 wild-type embryos. The kidney capsule was removed and the nephrogenic zone digested by incubating kidneys in collagenaseA/pancreatin mix for 15 min

according to *Brown et al. (2015)*. Nephron progenitors were purified by MACS sorting and resuspended in Apel medium supplemented with 2% PFHM II (Protein Free Hybridoma Medium II), FGF9 (200 ng/ml) and Heparin (1 µg/ml) as growth control medium. Freshly isolated cells were seeded at 200,000 cells per 24-well dish and cultured for 48 hr at 37˚C, 5% $CO_2$. We tested the effect of medium supplementation with RSPO3 (200 ng/ml) + WNT3a (50 ng/ml) or with Chir (3 µM) only. After 2 days, cells were scrapped with RLT buffer (Qiagen) and RNA extracted using RNeasy Mini kit according to the manufacturer's instructions (Qiagen).

## Acknowledgements

We thank Clara Panzolini for technical assistance and the entire staff of the iBV animal facility for their dedication. We are grateful to Agnès Loubat (iBV Cytology platform, Nice) for valuable discussions and advices with cell culture settings. We are indebted to Hitoshi Okamoto (Riken Institute, Japan) for providing the *Rspo3^flox* allele and Thomas J Carroll, Andrew P McMahon and Antoine Reginensi for sharing in situ probes.

This work was supported by grants from the EC (EURenOmics Grant agreement 305608) and La Ligue Contre le Cancer (Equipe labelisée) and the Conseil général des Alpes Maritimes for the financing of the Intavis In situ Robot.

## Additional information

### Funding

| Funder | Grant reference number | Author |
|---|---|---|
| European Commission | 305608 | Andreas Schedl |
| Ligue Contre le Cancer | Equipe labelisee | Andreas Schedl |

The funders had no role in study design, data collection and interpretation, or the decision to submit the work for publication.

### Author contributions

Valerie PI Vidal, Conceptualization, Data curation, Formal analysis, Supervision, Validation, Investigation, Visualization, Methodology, Writing - original draft, Writing - review and editing; Fariba Jian-Motamedi, Investigation, Methodology, Writing - review and editing; Samah Rekima, Data curation; Elodie P Gregoire, Data curation, Formal analysis, Investigation, Visualization; Emmanuelle Szenker-Ravi, Marc Leushacke, Bruno Reversade, Resources, Investigation, Writing - review and editing; Marie-Christine Chaboissier, Conceptualization, Resources, Supervision, Project administration, Writing - review and editing; Andreas Schedl, Conceptualization, Resources, Formal analysis, Supervision, Funding acquisition, Investigation, Visualization, Writing - original draft, Project administration, Writing - review and editing

### Author ORCIDs

Valerie PI Vidal (ID) https://orcid.org/0000-0001-8385-835X
Bruno Reversade (ID) http://orcid.org/0000-0002-4070-7997
Marie-Christine Chaboissier (ID) http://orcid.org/0000-0003-0934-8217
Andreas Schedl (ID) https://orcid.org/0000-0001-9380-7396

### Ethics

Animal experimentation: The experiments described in this paper were carried out in compliance with the French and international animal welfare laws, guidelines and policies and were approved by the local ethics committee (PEA No NCE-2014-207 and PEA No: 2018060516474844 (V2)).

### Decision letter and Author response

Decision letter https://doi.org/10.7554/eLife.53895.sa1
Author response https://doi.org/10.7554/eLife.53895.sa2

## Additional files

### Supplementary files

• Supplementary file 1. Key resources table.

• Transparent reporting form

### Data availability

All data generated or analysed during this study are included in the manuscript and supporting files. Source data files have been provided for Figures 2B, 2C, 2E, 3C, 3D, 4E, 5D, 5F 6C, and Figure 1—figure supplement 1A.

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
