## [Decision Letter]

**Acceptance summary:**

The topic of the maintenance and differentiation of renal stem cells is of prime interest. The work examines the role of R-spondins in controlling nephron progenitors/stem cells and the formation of nephrons. With different Cre drivers, distinct roles for *Rspo1* and *Rspo3* in the formation of nephrons are suggested. This indicates that there are additional signals in the process of MET. The actions of R-spondins seem to be independent of *Lgr4/5/6* signalling. The results have implications for renal development and possibly for regenerative medicine in kidney disease.

**Decision letter after peer review:**

Thank you for submitting your article "Paracrine and autocrine R-spondin signalling is essential for the maintenance and differentiation of renal stem cells" for consideration by *eLife*. Your article has been reviewed by three peer reviewers, and the evaluation has been overseen by a Roel Nusse as Reviewing Editor and Kathryn Cheah as the Senior Editor. The following individual involved in review of your submission has agreed to reveal their identity: Melissa H Little (Reviewer #3).

The reviewers have discussed the reviews with one another and the Reviewing Editor has drafted this decision to help you prepare a revised submission.

The work examines the role of R-spondins in controlling nephron progenitors/stem cells and the formation of nephrons. With different Cre drivers, distinct roles for *Rspo1* and *Rspo3* in the formation of nephrons are suggested. This indicates that there are additional signals in the process of MET. The actions of R-spondins seem to be independent of *Lgr4/5/6* signalling.

Summary:

The reviewers have mixed opinions about the quality of the work but they converge on a number of important issues to be addressed.

Essential revisions:

1) Clarify the use of the term stem cells or progenitors.

2) A better quantification of the data is required: e.g. on gene expression, proliferation and apoptosis, pSMAD1/5.

3) Provide an explanation of the finding that Lgrs are not required for the Rspo phenotypes, and if there is no explanation, consider to leave the data out.

4) To address proliferative versus a differentiative roles by analysing the effects of R-spondins on freshly cultured NPCs.

Reviewer #1:

Vidal et al. are interested in signals responsible for the maintenance and differentiation of stem cells in the kidney. They come to the conclusion that RSPO1 and RSPO3, signals that amplify Wnt signaling, are redundant and permit WNT/β-catenin signalling. Deleting the genes for RSPO1 and RSPO3 leads to a loss of renal stem cells, while deletion of the genes in cap mesenchymal cells blocks MET together with loss of Bmp7 signaling. Mutations in LGR4/5/6, has a limited effect on progenitor numbers. The authors suggest that R-spondins have a role in permitting stem cell maintenance.

The data and the paper have some merit but in aggregate, the work is presented in a confusing manner and raises more questions than answers.

For one, there is the use of the term stem cells in the title and the Abstract. Stem cells are defined as self-renewing cells throughout life, but the studies here are limited to embryonic or fetal development of the kidney. The term progenitor is more appropriate, which is used as well by the authors but in an intermittent way raising confusion on what they intend to say.

A major problem is the lack of robust quantification of gene expression. Transcripts are detected by RNA in situ hybridization, using RNAscope technology. But how the number or size of the dots detected correlate to real differences in number of RNA molecules is not clear.

Why certain drivers of Cre are used in certain experiments is not clear. The *CAGG:CreER^TM^* driver in subsection “RSPO1 and RSPO3 are essential to maintain the pool of kidney progenitors” is ubiquitous, would eliminate Rspo expression everywhere, not only in the kidney. Are there effects on other organs? Secondary effects on the kidney? In subsection “Stromal *Rspo3* maintains nephron progenitors during late stages of kidney development” there is a *Foxd1-Cre* driver, which is not inducible.

The observation that a triple knockout of LGR4/5/6 has a very limited phenotype has been made before by two of the co-authors (Szenker-Ravi, Reversade et al.). The underlying mechanism is still not clear, and the current manuscript doesn't shed light on it either. In the absence of an explanation, it is questionable whether these data belong in the manuscript.

Reviewer #2:

The authors have addressed the role of R-spondins in proliferation and differentiation of nephron progenitor cells. Nephron progenitor cells exist in a discrete anatomic site in the developing kidney and both self renew throughout fetal life and give rise to nephrons. No NPCs exist in adult life. While the Wnt system has been thoroughly investigated in renal development direct relevance to human kidney malformations has been rarely suggested (PMID: 23520208). Importantly, R-spondins have been recently used to expand organoid cultures of adult kidney epithelia (Nat Biotech 2019). The cell types in adult cultures include proliferating mature epithelia and maybe "stem states" of mature epithelia but surely they do not include NPCs.

The manuscript is overall elegant and the experiments are properly performed. Nevertheless, the verdict of proliferation versus differentiation of NPCs following an R-spondin cue is still out there. For the manuscript to be acceptable to *eLife* I would suggest a major revision and additional experiments on NPC cultures. Currently R-spondins are not required to propagate NPC ex vivo. To make a strong case for a proliferative versus a differentiative roles , isolation of NPC and analysis of the effects of R-spondins on freshly cultured NPCs is critically needed. I would like to see these data.

Reviewer #3:

This study takes a close look at the role of R-spondins in the regulation of nephron progenitors and the subsequent formation of nephrons. Using a series of different Cre drivers specific to subsets of cells within the developing kidney, they identify distinct roles for *Rspo1* and *Rspo3* in the formation of nephrons during kidney development. This suggests the involvement of an additional signal to the previously described role of canonical Wnt signalling in the process of MET. Surprisingly, they also demonstrate that the actions of R-spondins are independent of Lgr signalling as the phenotypes were not replicated in cell-specific LGR knockouts. While the mechanism of action was not further investigated, the study was performed well and adds clarity to our understanding of the molecular basis of kidney development. The data presented in large part supports the conclusions that have been made by the authors. The quality of the data is generally high.

---

## [Author Response]

Essential revisions:1) Clarify the use of the term stem cells or progenitors.

The term “renal stem cell” has been used repeatedly in the literature for nephron precursors, but we agree that nephron progenitors are less controversial. We have therefore replaced stem cells with nephron progenitors throughout the manuscript.

2) A better quantification of the data is required: e.g. on gene expression, proliferation and apoptosis, pSMAD1/5.

Proliferation and apoptosis:

We have noticed that the descriptions provided in our original submission on how quantifications were performed were incomplete and we would like to apologize for this oversight. We have now revised the Materials and methods section and also provide more details on the number of samples analyzed (n) in the figure legends.

Gene expression:

Many of the genes analysed show a highly dynamic expression pattern in kidney development and their expression is not restricted to the nephron progenitor compartment. This makes quantification by qPCR unsuitable. We had therefore opted for in situ hybridization analysis that provides spatial information, and a qualitative assessment of RNA expression. RNA-Scope analysis is considered more quantitative when counting individual dots, but cost-prohibitive if a large number of genes have to be analysed. As a compromise, we have now performed new RNA-Scope experiments for the direct β-catenin target *Axin2* and quantified its expression using the HALO software package. Our new data demonstrate a significant reduction of *Axin2* expression upon NPC-specific deletion of R-spondins (Figure 4E and Figure 2—figure supplement 1). The use of the Halo Software required the expertise of Samah Rekima (plateform Histology, iBV), who has been added as a co-author in the revised manuscript.

pSMAD1/5:

We have now quantified pSMAD1/5 cells within the nephrogenic zone on 4 independent knockout and wildtype samples. Our data confirm a >90% reduction for SMAD1/5 activation and this data have now been added as Figure 5F.

3) Provide an explanation of the finding that Lgrs are not required for the Rspo phenotypes, and if there is no explanation, consider to leave the data out.

LGR4-6 are considered as the main receptors of R-spondins (de Lau 2012). However, recent results demonstrate that RSPO3 can also act through an LGR-independent mechanism involving HSPGs (Lebensohn and Rohatgi, 2018). As discussed in our manuscript, the *Lgr* knockout kidney phenotype is far less severe than that observed in R-spondins mutants, which implies that the latter must act in an LGR independent manner. Providing functional proof that they act via HSPGs (or through alternative adaptors) is difficult and beyond the scope of this paper. While LGRs only play a minor role in mediating R-spondin activity in the kidney, we believe this observation is still important. If the reviewers insist, we could eventually move the data to the supplementary figures section.

4) To address proliferative versus a differentiative roles by analysing the effects of R-spondins on freshly cultured NPCs.

Depending on culture conditions isolated NPCs can be either expanded (low density plating) or induced to undergo MET (high density plating with a pulse of β-catenin inducing agent). We had planned to evaluate the effect of recombinant R-spondins under both of these conditions.

Low density plating:

We used the previously established medium by Brown et al., 2015, and cultured freshly isolated NPCs for 72h replacing Chiron (or not) with RSPO3. Surprisingly, in our hands, NPC expansion was also achieved in the absence of Chiron (and RSPO3), perhaps because intrinsic factors (cells continue to produce R-spondins) were sufficient to ensure expansion for the first 72h. Additional experiments, using R-spondin knockout mice had to be abandoned because of the COVID19 lockdown. However, deletion of *Rspo1*/3 at any time point during kidney development (we tested E10.5/E11.5/E12.5/ E13.5) leads to a rapid decline of the NPC population (Figure 2), which clearly demonstrates that these signalling molecules are essential for maintaining the progenitor pool.

High density plating:

Kidney progenitors were freshly isolated according to Brown et al., 2015, and cultured for 48h under high density plating conditions in the presence of RSPO3 (200μg/ml) or RSPO3 (200μg/ml) + Wnt3a (50ng/ml) or with Chiron (3μM). Under those conditions recombinant RSPO3 appears to induce canonical β-catenin signalling and initiate the initial steps of MET, as evidenced by the upregulation of *Axin2* and *Wnt4*. (Figure 5D) (unfortunately we could not test other markers due to the COVID-19 lockdown). These new data have been included in the revised manuscripts.

Reviewer #1:Vidal et al. are interested in signals responsible for the maintenance and differentiation of stem cells in the kidney. They come to the conclusion that RSPO1 and RSPO3, signals that amplify Wnt signaling, are redundant and permit WNT/β-catenin signalling. Deleting the genes for RSPO1 and RSPO3 leads to a loss of renal stem cells, while deletion of the genes in cap mesenchymal cells blocks MET together with loss of Bmp7 signaling. Mutations in LGR4/5/6, has a limited effect on progenitor numbers. The authors suggest that R-spondins have a role in permitting stem cell maintenance.The data and the paper have some merit but in aggregate, the work is presented in a confusing manner and raises more questions than answers.For one, there is the use of the term stem cells in the title and the Abstract. Stem cells are defined as self-renewing cells throughout life, but the studies here are limited to embryonic or fetal development of the kidney. The term progenitor is more appropriate, which is used as well by the authors but in an intermittent way raising confusion on what they intend to say.

We have replaced the term renal stem cells with nephron progenitors throughout the text.

A major problem is the lack of robust quantification of gene expression. Transcripts are detected by RNA in situ hybridization, using RNAscope technology. But how the number or size of the dots detected correlate to real differences in number of RNA molecules is not clear.

Please see our response to point 2 of the Essential revisions section.

Why certain drivers of Cre are used in certain experiments is not clear. The CAGG:CreERTM driver in subsection “RSPO1 and RSPO3 are essential to maintain the pool of kidney progenitors” is ubiquitous, would eliminate Rspo expression everywhere, not only in the kidney. Are there effects on other organs? Secondary effects on the kidney? In subsection “Stromal Rspo3 maintains nephron progenitors during late stages of kidney development” there is a Foxd1:Cre driver, which is not inducible.

We apologize for the confusion. The inducible ubiquitous driver (*CAGG:CreER^TM^*) has been used to efficiently delete R-spondins in all cell types. Since the effect on kidney progenitor proliferation and survival is very rapid (48h) and can be seen independently of the time of induction (we tested E9.5, E10.5, E11.5, E12.5, E13.5), the phenotype is unlikely to be due to secondary effects caused by phenotypes in other organ systems. Moreover, R-spondins are considered to work locally rather than on an organismal level. Finally, loss of progenitor cell maintenance is also seen when using tissue specific Cre lines such as *Foxd1-Cre* (see Figure 3). We have modified the text to better explain why certain drivers have been used.

The observation that a triple knockout of LGR4/5/6 has a very limited phenotype has been made before by two of the co-authors (Szenker-Ravi, Reversade et al.). The underlying mechanism is still not clear, and the current manuscript doesn't shed light on it either. In the absence of an explanation, it is questionable whether these data belong in the manuscript.

The paper by Szenker-Ravi et al. did not report on the kidney phenotype and we believe that the information that RSPO signalling in the kidney only partially depends on LGR receptors is important to the scientific community. We would therefore prefer to maintain this information in the paper. If the editors feel strongly against the inclusion as a main figure, we could move these data to the supplementary section. (please see also our response to point 3 of the Essential revisions section.)

Reviewer #2:The authors have addressed the role of R-spondins in proliferation and differentiation of nephron progenitor cells. Nephron progenitor cells exist in a discrete anatomic site in the developing kidney and both self renew throughout fetal life and give rise to nephrons. No NPCs exist in adult life. While the Wnt system has been thoroughly investigated in renal development direct relevance to human kidney malformations has been rarely suggested (PMID: 23520208). Importantly, R-spondins have been recently used to expand organoid cultures of adult kidney epithelia (Nat Biotech 2019). The cell types in adult cultures include proliferating mature epithelia and maybe "stem states" of mature epithelia but surely they do not include NPCs.The manuscript is overall elegant and the experiments are properly performed. Nevertheless, the verdict of proliferation versus differentiation of NPCs following an R-spondin cue is still out there. For the manuscript to be acceptable to eLife I would suggest a major revision and additional experiments on NPC cultures. Currently R-spondins are not required to propagate NPC ex vivo. To make a strong case for a proliferative versus a differentiative roles , isolation of NPC and analysis of the effects of R-spondins on freshly cultured NPCs is critically needed. I would like to see these data.

Please see our response to point 4 of the Essential revisions section.